# Interleukin-10-Mediated Lymphopenia Caused by Acute Infection with Foot-and-Mouth Disease Virus in Mice

**DOI:** 10.3390/v13122358

**Published:** 2021-11-24

**Authors:** Zijing Guo, Yin Zhao, Zhidong Zhang, Yanmin Li

**Affiliations:** 1State Key Laboratory on Veterinary Etiological Biology, Lanzhou Veterinary Research Institute, Chinese Academy of Agricultural Sciences, Lanzhou 730030, China; zijingguo7@163.com (Z.G.); zylsy2019@126.com (Y.Z.); 2College of Animal Husbandry and Veterinary Medicine, Southwest Minzu University, Chengdu 610041, China

**Keywords:** foot-and-mouth disease virus, lymphopenia, interleukin-10, apoptosis, trafficking, co-inhibitory molecules

## Abstract

Foot-and-mouth disease (FMD) is characterized by a pronounced lymphopenia that is associated with immune suppression. However, the mechanisms leading to lymphopenia remain unclear. In this study, the number of total CD4^+^, CD8^+^ T cells, B cells, and NK cells in the peripheral blood were dramatically reduced in C57BL/6 mice infected with foot-and-mouth disease virus (FMDV) serotype O, and it was noted that mice with severe clinical symptoms had expressively lower lymphocyte counts than mice with mild or without clinical symptoms, indicating that lymphopenia was associated with disease severity. A further analysis revealed that lymphocyte apoptosis and trafficking occurred after FMDV infection. In addition, coinhibitory molecules were upregulated in the expression of CD4^+^ and CD8^+^ T cells from FMDV-infected mice, including CTLA-4, LAG-3, 2B4, and TIGIT. Interestingly, the elevated IL-10 in the serum was correlated with the appearance of lymphopenia during FMDV infection but not IL-6, IL-2, IL-17, IL-18, IL-1β, TNF-α, IFN-α/β, TGF-β, and CXCL1. Knocking out IL-10 (IL-10^-/-^) mice or blocking IL-10/IL-10R signaling in vivo was able to prevent lymphopenia via downregulating apoptosis, trafficking, and the coinhibitory expression of lymphocytes in the peripheral blood, which contribute to enhance the survival of mice infected with FMDV. Our findings support that blocking IL-10/IL-10R signaling may represent a novel therapeutic approach for FMD.

## 1. Introduction

FMD is recognized as one of the most infectious and economically important diseases of domestic livestock [1]. The etiological agent, FMD virus (FMDV), an *aphthovirus* of the *Picornaviridae* family, can infect a multitude of cloven-hoofed animal species, including both ruminants and suids [1,2]. The FMDV genome is approximately 8.5 kb in length, with a positive-sense, single-stranded RNA that encodes four structural proteins (VP1, VP2, VP3, and VP4) and eight nonstructural proteins (L, 2A, 2B, 2C, 3A, 3B, 3C, and 3D) [1]. To date, seven known serotypes of FMDV (A, O, Asia1, C, SAT1, SAT2, and SAT3) exist and consist of numerous subtypes [1].

Following FMDV infection, lymphocyte depletion in the peripheral blood, referred to as lymphopenia, is a common characteristic in pigs [3,4], cattle [5,6], and C57BL/6 mice [7]. Lymphopenia is considered to be one of the important mechanisms by which the FMDV evade the host immune response and cause immune suppression [2,8]. However, the mechanisms of lymphopenia caused by FMDV are still unclear. A previous study reported that FMDV serotype C can directly infect T and B cells in lymphoid compartments of swine, which may result in lymphoid depletion [4]. Despite high viremia in pigs infected with FMDV (serotypes of C, O, and A), no viral RNA or virus could be isolated on BHK-21 cells from the lysate of peripheral blood mononuclear cells (PBMCs), indicating that the active infection of lymphocytes is not believed to be responsible for lymphopenia [3,9]. Thus, it was hypothesized that lymphopenia may be induced by the tissue redistribution of lymphocyte and/or apoptosis through the activation of caspase 3, as IFN-α/β of Type 1 interferons (Type 1 IFN), reported to be related to the trafficking of lymphocytes, was increased following FMDV infection, and the expression of caspase 3 mRNA was upregulated more in PBMC from infected pigs compared to uninfected controls [3,10,11]. Until now, no direct evidence of peripheral blood lymphocyte apoptosis and trafficking have been reported following FMDV infection.

Interleukin-10 (IL-10), a master regulator of the anti-inflammatory response, is critical to protect the host from tissue damage during acute phases of immune responses [12]. IL-10 can be produced by virtually all immune cells, and it can also modulate the function of these cells [13]. Despite its primary anti-inflammatory role, IL-10 plays a dual role of promoting pathogen persistence [14,15,16,17] and limiting immune pathology [18,19], and these effects may be due to the IL-10-producing immune cell, the available IL-10 levels, the target cell, and the microenvironmental cofactors, such as other cytokines [20]. It has been suggested that IL-10 was able to suppress T-cell proliferation [12]. In animal models of chronic virus infection, blocking IL-10 signaling could successfully prevent T-cell exhaustion [15]. Furthermore, IL-10 produced by CD9^+^-regulatory B cells was found to induce T-cell apoptosis [21], which indicated that IL-10 may be involved in lymphopenia. During FMDV infection, the elevated IL-10 level has been widely reported in the serum of FMDV-infected cattle and pigs, with a peak at day 3 or 4 post-infection, which coincides with the development of the clinical disease [22,23,24]. The elevated IL-10 produced by dendritic cells (DCs) suppresses T-cell proliferation during FMDV infection [22,25]. In addition, it also may impair the major histocompatibility complex class II molecule (MHC II) expression and antigen processing function during FMDV infection [23]. Thus, elevated IL-10 has been proposed to be related to immune suppression during FMDV infection [22,26]. However, it is still unclear if IL-10 is involved in lymphopenia caused by FMDV infection.

In the present study, we reported that lymphopenia caused by FMDV was associated with the disease severity, and IL-10 was related to the occurrence of lymphopenia caused by FMDV infection. IL-10^-/-^ or blocking IL-10/IL-10R signaling in vivo is able to prevent lymphopenia via downregulating apoptosis, trafficking, and the expression of coinhibitory molecules of lymphocytes from FMDV-infected mice, which contribute to enhance the survival of mice infected with FMDV. These results could provide new insights into the pathogenicity of FMDV and potential therapeutics for FMDV-infected animals.

## 2. Materials and Methods

### 2.1. Mice and Virus Strain

C57BL/6 mice at age of four to five weeks were obtained from the Animal Center for Lanzhou Veterinary Research Institute (LVRI), Chinese Academy of Agricultural Sciences (CAAS). C57BL/6-background IL-10^-/-^ mice were purchased from Cyagen Biosciences Inc (Guangzhou, China). All infectious animal work was performed in the biosafety level 3 biocontainment laboratory at LVRI, in strict accordance with good animal practice as stipulated by the Animal Ethics Procedures and Guidelines of the People’s Republic of China, and the study was approved by the Animal Ethics Committee of LVRI, CAAS (Approval Code: LVRIAEC-2019-085, approval date: 20 December 2019). The animals were euthanized when infected mice showed severe respiratory distress, inability to maintain body temperature, or secondary infection. FMDV O/BY/CHA/2010 (GenBank accession number JN998085) was used for viral infection experiments in this study. O/BY/CHA/2010 was obtained from the National/OIE Foot-and-Mouth Diseases Reference Laboratory, LVRI, CAAS.

### 2.2. Animal Infection and Preparation of Samples

Mice were infected with FMDV subcutaneously (0.05 mL) with 5 × 10^5.5^ TCID50 per mice, which was the lowest dose of infection previously determined by our laboratory to cause death in mice at 4-6 weeks [7]. Mock mice were inoculated subcutaneously with 0.05-mL phosphate-buffered saline (PBS). C57BL/6 mice infected with FMDV developed clear signs of disease, including respiratory distress, neurological signs, and wasting [27]. At 48 h post-inoculation (hpi), mice were euthanized, and tissues of the liver, lung, spleen, heart, and kidney were collected and stored at −80 °C until use. The serum was isolated from the whole blood obtained from the infraorbital sinus of the mice at 12, 24, 36, 48, and 60 hpi. Blood was collected in tubes containing an ethylene diamine tetra acetic acid (EDTA) anticoagulant, and the PBMCs were isolated using density gradient centrifugation (Lymphocyte Separation Medium; Tianjin Haoyang Inc., Tianjin, China). The PBMCs were resuspended in staining buffer (PBS containing 2% fetal bovine serum) for flow cytometry. Before sample collection, the mice were sacrificed using intraperitoneal (i.p.) injections with 200-μL pentobarbital, followed by cervical dislocation or exsanguination under terminal anesthesia.

### 2.3. Antibodies and Flow Cytometry

Anti-mouse CD3, CD19, NK1.1, CD8, and CD4 were used to identify T lymphocytes, B lymphocytes, natural killer (NK) lymphocytes, CD3^+^CD8^+^ T lymphocytes, and CD3^+^CD4^+^ T lymphocytes, respectively. Anti-mouse PD-1, CTLA-4, Tim-3, LAG-3, 2B4, TIGIT, CD160, and the isotype control of these antibodies were used to evaluate CD4^+^ and CD8^+^ T cell exhaustion. Anti-mouse CD16/32 was used to preincubate for blocking Fc receptors. The detailed information of these antibodies was shown in Appendix A. For cell surface staining, antibodies were incubated for 20 min on the ice in staining buffer. After staining, all samples were washed by PBS twice. Lymphocytes populations were selected on the basis of forward and side light scatter and counting the number of lymphocytes by flow cytometry. Data were acquired on a CytoFLEX (Beckman Coulter Inc., Shanghai, China) and were analyzed with FlowJo software (Treestar, Ashland, OR, USA).

### 2.4. In Vivo Antibody Treatment

In vivo anti-mouse IL-10R (clone 1B1.3A) for blocking IL-10/IL-10R signaling and in vivo rat IgG1 isotype control (clone HRPN) were purchased from BioXCell (West Lebanon, NH, USA). Mice were injected with an intraperitoneal injection (i.p.) with 200 μg of anti-IL-10R Ab (clone 1B1.3A) or a control rat IgG1 Ab (clone HRPN), as described previously [28], on days 1 and 0 after FMDV infection.

### 2.5. Apoptosis Detection

The PBMCs were isolated from mock mice and FMDV-infected mice. The cells were labeled separately with surface markers, Alexa Flour@ 647 anti-mouse CD3, APC anti-mouse CD19, and APC anti-mouse NK1.1 in staining buffer for 20 min on ice in the dark to detect the apoptosis of T, B, and NK cells. After washing, cells apoptosis was detected using the Annexin V/PI detection kit (BD Biosciences Inc., Franklin Lakes, N.J., USA) according to the manufacturer’s instructions. Annexin V+/PI− cells were considered early apoptotic cells, and Annexin V+/PI+ cells were considered late apoptotic cells [29] using CytoFLEX (Beckman Coulter Inc., Shanghai, China).

### 2.6. RT-qPCR

The serums were collected from mock mice and infected mice at 12, 24, 36, 48, and 60 h post-FMDV infection; the total RNAs were extracted from 100-μL serum using TRIzol reagent (Invitrogen, Waltham, MA, USA). The cDNA was synthesized from the extracted RNA preparation using PrimeScript™ RT Master Mix (Takara, Tokyo, Japan), according to the manufacturer’s instructions. FMDV RNA in the serums was quantitatively determined by RT-qPCR, and the primers targeting the FMDV 3D gene were reported in a previous study [30].

Tissues samples of the heart from mock mice or infected mice were collected at 48 hpi. The total RNAs were extracted using TRIzol reagent (Invitrogen, Waltham, MA, USA). One microgram of purified RNA was transcribed into cDNA using the PrimeScript™ RT reagent kit with gDNA Eraser (Takara, Tokyo, Japan) according to the manufacturer’s instructions. The levels of the lymphocytes in the heart were analyzed by RT-qPCR using TB Green Premix ExTaq reagents (TaKaRa, Tokyo, Japan) and with the following forward (F) and reverse (R) primers. The primer sequences were as follows: CD3-F: GCCCAGAGGGCAAAACAAG, CD3-R: TGCGGATGGGCTCATAGTCT. NK1.1-F: GACTCTCCCGAAACCCATCA, and NK1.1-R-GTTCACCGAGTTTCCATTTGTG. The GAPDH (Mm01205647_g1) and CD19 (Mm01205647_g1) primers were purchased from Life Technologies (Shanghai, China). The relative expression of mRNA was calculated using the comparative cycle threshold (2–^ΔΔCT^) method. All the experiments were repeated three times for each experiment.

### 2.7. Tracking Lymphocytes Labeled with CFSE In Vivo

Carboxyfluorescein diacetate succinimidyl ester (CFSE) can be used to label a cell population of interest easily and quickly for in vivo investigation. This labeling has classically been used to study lymphocyte proliferation and trafficking [31]. Splenic lymphocytes were isolated by the Mouse Spleen Lymphocyte Isolation Kit (Tianjin Haoyang Inc., Tianjin, China), and these lymphocytes were labeled CFSE with the CellTrace™ Cell Proliferation Kit (Invitrogen, Waltham, MA, USA) according to the manufacturer’s instructions. These cells were resuspended in 0.9% NaCl and the numbers counted again, and the staining was assessed by a FACS analysis of the excess cells prior to injection. The suspension of 5 × 10^6^-labeled cells was injected into the tail veins of per mice before the virus challenge. After this, the animals were classified as the virus-infected group and mock group, with three mice in each group. The animals were killed at 48 hpi and dissected as described previously for the CFSE-labeled lymphocytes analysis. The lymphocytes from the spleen and blood were analyzed directly with the flow cytometer gated on the lymphocyte populations, counting 20,000 lymphocytes for each compartment. The heart, lung, liver, and kidney were observed by fluorescence microscope.

### 2.8. Detection of Cytokine Levels in the Serum

The IL-10, IL-6, tumor necrosis factor-α (TNF-α), IFN-α, IFN-β, transforming growth factor-β (TGF-β), and CXCL13 enzyme linked immunosorbent assay (ELISA) kits were purchased from MyBioSource (Vancouver, DC, USA), while the IL-2, IL-17, IL-18, IL-1β, and CXCL1 ELISA kits were purchased from Cloud Clone Corp (Wuhan, China). The level of cytokines in the serum was measured according to the manufacturer’s instructions.

### 2.9. Statistical Analysis

All statistical analyses were performed using GraphPad Prism version 5.03 software. The cell counts, the expression of coinhibitory molecules, and cytokine/chemokines levels were compared by an unpaired *t*-test. The percent of T, B cell, CD4+, and CD8+ T cells and the apoptosis and the mRNA expression of T, B, and NK cells were analyzed by two-way ANOVA with Tukey’s multiple comparison test. The survival rates in the mice were compared by the log-rank test. Throughout this article, *, **, ***, and **** indicate statistically significant differences, with *p*-values of <0.05, 0.01, 0.001, and 0.0001 respectively, while ns indicates a non-statistically significant difference (*p* > 0.05).

## 3. Results

### 3.1. FMDV Infection Was Lethal to C57BL/6 Mice

All infected mice (*n* = 15) showed signs of clinical disease at 36–72 h post-inoculation (hpi) and died within 4–6 h after showing signs of severe clinical disease. The infected mice initially showed lethargy, isolation, and anorexia, which were scored as mild clinical symptoms in this study. After that, the infected mice showed respiratory distress, ruffled fur, and humped posture, which were scored as severe clinical symptoms in this study. The quick disease progression resulted in the deaths of all animals within 72 hpi (Appendix A). To investigate the kinetics of the virus in the serum of the infected mice, time course studies were carried out, and the viral RNA level in the serum was determined at different times post-infection. The FMDV RNA (1.1 ± 1.8 × 10^8^ copies viral RNA/100-μL serum, *n* = 3) was detected in infected mice as early as 36 hpi, and the highest level was 5.9 × 10^8^ copies viral RNA/100-μL serum at 60 hpi with the severe clinical symptoms (Figure 1D). It is noted that the signs of clinical disease were observed coinciding with the onset of FMDV RNA in the serum detected.

### 3.2. Lymphopenia in Mice Acutely Infected with FMDV

The number of lymphocytes in the peripheral blood was monitored in mice infected with type O FMDV (5 × 10^5.5^ TCID50) by using flow cytometry. The results showed that a significant reduction in the number and percentage of lymphocytes counts occurred at 36 (120.3 ± 0.1 × 10^4^ cells/mL, *n* = 3, *p* = 0.0016) and then further decreased at 48 and 60 hpi (12.0 ± 0.3 × 10^4^ and 8.5 ± 1.3 × 10^4^ cells/mL, respectively, *n* = 3, *p* < 0.0001) when compared with that of mock mice (213.4 ± 0.1 × 10^4^ cells/mL, *n* = 3) (Figure 1A,B). Notably, the infected mice with severe clinical symptoms showed significantly lower lymphocyte counts (6.2 ± 1.6 × 10^4^ cells/mL, *n* = 5) than those mock mice (202.6 ± 8.7 × 10^4^ cells/mL, *n* = 5, *p* < 0.0001) or the infected mice without severe clinical symptoms (94.7 ± 17.1 × 10^4^cells/mL, *n* = 5, *p* = 0.0009) (Figure 1C), which indicated that the lymphocyte count was associated with the disease severity in mice infected with FMDV. Indeed, the drop in lymphocyte counts coincided with the onset of FMDV RNA in the serum of the infected mice (Figure 1B,D). The viral RNA loads in the serum were detected in the infected mice at 36 hpi, and they were significantly increased in the infected mice at 48 hpi and continued to increase at 60 hpi (Figure 1D). Overall, the infected mice with severe clinical symptom showed obvious lymphopenia and high viral RNA loads in the serum.

### 3.3. Loss of CD3+, CD19+, and NK1.1+ Cells Involved in Lymphopenia

In order to determine which lymphocyte subset counts were reduced during FMDV infection in mice, PBMCs were isolated from mock and FMDV-infected mice at different times post-infection. The PBMCs obtained were then incubated with anti-mouse CD19, anti-mouse NK1.1, anti-mouse CD3, anti-mouse CD4, and anti-mouse CD8 antibodies and analyzed by flow cytometry (Appendix A). The results showed that the number of CD3^+^ T cells, CD19^+^ B cells, and NK1.1^+^ cells were reduced during FMDV infection (Figure 2A–C). An obvious reduction in the number of CD3^+^ T cells and CD19+ B cells occurred from 48 hpi, while the NK1.1^+^ cells obviously decreased from 12 hpi (Figure 2C). A further analysis revealed that the percentages of the T cells and B cells in the peripheral blood changed with the times post-infection. The percentage of B cells was initially decreased at 12 hpi (mock vs. 12 hpi, *n* = 3, *p* = 0.9982) and then increased at 24 hpi (mock vs. 24 hpi, *n* = 3, *p* = 0.0147); after that, it gradually decreased from 36 to 60 hpi (mock vs. 48 hpi and mock vs. 60 hpi, *n* = 3, *p* = 0.0321 and 0.0436, respectively) (Figure 2D). Among the CD3^+^ T cells, the percentage of CD3+CD8+CD4- T cells decreased during the whole progression of FMDV infection (Figure 2E). Indeed, the percentage of CD19^+^ B cells in infected mice with severe clinical symptoms (10.5 ± 1.0%, *n* = 8, *p* < 0.0001) was significantly reduced when compared with mock mice (27.6 ± 2.5%, *n* = 8) (Figure 2F), and among CD3^+^ T cells, the percentage of CD3^+^CD8^+^CD4^−^ lymphocytes was significantly reduced in infected mice with severe clinical symptoms (26.4 ± 1.9%, *n* = 8, *p* = 0.0196) when compared with mock mice (34.9 ± 1.6%, *n* = 8) (Figure 2F). These results suggested that T cells, B cells, and NK cells were involved in lymphopenia, and the loss of B cells was more severe as compared to T cells; among the T cells, CD3^+^CD8^+^CD4^−^ T cells was the main loss of the T population.

### 3.4. Apoptosis of T Cells, B Cells, and NK Cells Induced following FMDV Infection

The rapid depletion of lymphocytes in the blood could be due to virus-induced apoptosis cells. To determine whether peripheral blood lymphocytes suffered apoptosis following FMDV infection in mice, we applied annexin V^+^/PI^+^ among the total CD3^+^ T cells, CD19^+^ B cells, and NK1.1^+^ NK cells in the peripheral blood. The results showed that the apoptosis of CD3^+^ T cells was increased in FMDV-infected mice at 48 and 60 hpi (66.6 ± 6.7% and 66. 3 ± 5.9%, *n* = 3, *p* = 0.0018 and *p* = 0.0020, respectively) when compared with the mock mice (27.4 ± 0.5%, *n* = 3) and, among the late apoptosis of CD3^+^ T cells, was significantly increased in FMDV-infected mice at 48 and 60 hpi (47.1 ± 7.5% and 45.6 ± 6.6%, *n* = 3, *p* = 0.0007 and *p* = 0.0013, respectively) when compared with the mock mice (17 ± 0.5%, *n* = 3) (Figure 3A,D). The apoptosis of CD19^+^ B cells was also increased in FMDV-infected mice at 36 and 48 hpi (39.4 ± 1.8% and 41.3 ± 4.4%, *n* = 3, *p* = 0.0089 and *p* = 0.0033, respectively) when compared with the mock mice (21.0 ± 1.8%, *n* = 3), and the late apoptosis of CD19^+^ B cells was significantly increased in FMDV-infected mice at 48 hpi (29.0 ± 4.7%, *n* = 3, *p* = 0.0083) when compared with the mock mice (15.5 ± 2.1%, *n* = 3) (Figure 3B,E). In addition to 60 hpi, the apoptosis of NK cells was increased with time, and the apoptosis of NK cells was significantly greater in FMDV-infected mice at 48 hpi (39.2 ± 0.4%, *n* = 3, *p* = 0.0355) than that of the mock mice (18.2 ± 4.0%, *n* = 3) (Figure 3C,F). These results suggested that the apoptosis of T cells, B cells, and NK cells was induced following FMDV infection, especially T cells.

### 3.5. Trafficking of Lymphocytes from the Peripheral Blood to Heart following FMDV Infection

Lymphocytes being sequestered from the circulation and migration to infected tissues could lead to the rapid depletion of lymphocytes in the blood. To determine if lymphocytes migrated from the peripheral blood to infected tissue following FMDV infection, the mice were injected with the CFSE-labeled spleen lymphocytes intravenously, and the blood, spleen, heart, kidney, lung, and liver were collected at 48 hpi. The CFSE-labeled lymphocytes in the blood and spleen were detected by flow cytometry, and the CFSE-labeled lymphocytes in the heart, kidney, and lung were observed by fluorescence microscope, because it is difficult to separate lymphocytes from these tissues. The results showed that the CFSE-labeled lymphocytes in the blood of the infected mice (6.7 ± 2.0 × 10^2^ cells/mL, *n* = 3, *p* = 0.0016) were significant decreased compared with that of the mock mice (48.1 ± 5.4 × 10^2^ cells/mL, *n* = 3), which further supported lymphopenia induced by FMDV infection (Appendix A). Although the CFSE-labeled lymphocytes in the spleen of the infected mice (6.2 ± 1.1 × 10^4^ cells/mL, *n* = 3, *p* = 0.2646) were also reduced, there is no significant difference with the mock mice (8.0 ± 0.9 × 10^4^ cells/mL, n = 3) (Appendix A). By using an immunofluorescence microscope, the CFSE-labeled lymphocytes were observed in tissues of the heart, kidney, lung, and liver. The results showed that the levels of CFSE-labeled lymphocytes in the kidneys, lungs, and livers of the infected mice were similar to those of the mock mice (Figure 4). However, the CFSE-labeled lymphocytes in the hearts of the infected mice were obviously increased compared with that of the mock mice (Figure 4). Taken together, these results suggested that lymphocytes migrate from the peripheral blood to the heart following FMDV infection.

### 3.6. The Upregulated Expression of Coinhibitory Molecules on T Lymphocytes during FMDV Infection

The coinhibitory receptors such as PD-1, LAG-3, and Tim-3 are considered T cell exhaustion markers, and the upregulated expression of these coinhibitory molecules were reported to inhibit T-cell proliferation and then promote apoptosis. To determine whether the expression of coinhibitory molecules was upregulated during acute infection with FMDV in mice, the expression of all known seven coinhibitory molecules (PD-1, CTLA-4, Tim-3, LAG-3, 2B4, TIGIT, and CD160) was analyzed on peripheral blood T lymphocytes, which were harvested from the mock mice and infected mice at 48 hpi when the increased apoptosis of T lymphocytes occurred (Appendix A). The results showed that TIM-3 and CD160 expression on CD4+ and CD8+ T cells from FMDV-infected mice was not obviously affected during FMDV infection as compared to the mock mice (CD4-Tim3, 3.1 ± 0.4% vs. 2.4 ± 0.2% and CD8-Tim3, 1.9 ± 0.6% vs. 1.2 ± 0.2%, *n* = 10, *p* = 0.1165 and *p* = 0.2459, respectively; CD4-CD160, 1.6 ± 0.5% vs. 0.9 ± 0.1% and CD8-CD160, 10.4 ± 2.2% vs. 16.5 ± 2.0%, *n* = 10, *p* = 0.2621 and *p* = 0.0528, respectively) (Figure 5). However, CTLA-4, LAG-3, 2B4, and TIGIT were indeed highly expressed on both CD4+ and CD8+ T cells from FMDV-infected mice, as compared to the mock mice (CD4-CTLA4, 4.3 ± 0.7% vs. 2.2 ± 0.3% and CD8-CTLA4, 2.4 ± 0.5% vs. 1.2 ± 0. 2%, *n* = 10, *p* = 0.0169 and *p* = 0.0457, respectively; CD4-LAG3, 1.8 ± 0.3% vs. 1.0 ± 0.2% and CD8-LAG3, 2.5 ± 0.4% vs. 1.2 ± 0.1%, *n* = 10, *p* = 0.0352 and *p* = 0.0026, respectively; CD4-2B4, 1.5 ± 0.2% vs. 0.9 ± 0.1% and CD8-2B4, 11.6 ± 1.6% vs. 3.0 ± 0.3%, *n* = 10, *p* = 0.0235 and *p* < 0.0001, respectively; CD4-TIGIT, 4.1 ± 0.7% vs. 1.7 ± 0.1% and CD8-TIGIT, 4.9 ± 0.6% vs. 2.9 ± 0.3%, *n* = 10, *p* = 0.0150 and *p* = 0.0167, respectively); the PD-1 on CD8^+^ T cells from FMDV-infected mice also displayed a high expression, as compared to the mock mice (12.0 ± 1.4% vs. 7.2 ± 0.5%, *n* = 10, *p* = 0.0056) (Figure 5). These results showed that the expression of multiple coinhibitory molecules was upregulated on CD4^+^ and CD8^+^ cells during FMDV infection.

### 3.7. The Elevated Levels of IL-10 Coincided with Development of Lymphopenia following FMDV Infection

As per an earlier report, the elevated IL-10 level was detected in the serum of FMDV-infected cattle and pigs, which coincided with the development of the clinical disease. To determine whether IL-10 was upregulated following FMDV infection in mice and then explore its association with the lymphopenia observed, we examined the concentrations of IL-10 and other potential cytokines/chemokines, including IL-10, IL-6, IL-2, IL-17, IL-18, IL-1β, TNF-α, IFN-α, IFN-β, TGF-β, CXCL13, and CXCL1, in the serum from the mice infected with FMDV. As shown in Figure 6, only the levels of IL-10 and CXCL-13 were significantly increased in FMDV-infected mice, especially in the infected mice with severe clinical symptoms. Importantly, the level of IL-10 was significantly increased from 48 hpi (16.5 ± 4.1 pg/mL, *n* = 3, *p* = 0.0266) and then continued to increase until the deaths of the infected mice at 72 hpi (57.6 ± 4.0 pg/mL, *n* = 3, *p* = 0.0002) when compared with the mock mice (2.3 ± 0.7 pg/mL, *n* = 3), which not only coincided with the onset of the viral RNA level in the serum but also coincided with the onset of lymphopenia (Appendix A). The elevated levels of IL-10 also coincided with the elevated levels of CXCL13 (Figure 6). In contrast, the levels of IL-6, IL-2, IL-17, IL-18, IL-1β, TNF-α, IFN-α/β, TGF-β, and CXCL1 in the serum of infected mice were not different from those of the mock mice (Figure 6).

### 3.8. Blocking IL-10 Signaling with IL-10 Receptor Blocker Prevented Lymphopenia Driven by FMDV Infection

IL-10 was previously reported to induce T-cell apoptosis [21]. Therefore, to determine if IL-10 mediated apoptosis, leading to the decline of peripheral blood lymphocytes in FMDV-infected mice, IL-10R blocker for blocking IL-10/IL-10R signaling was used to inject into the mice on days 1 and 0 after FMDV infection. The survival time of mice and the number of peripheral blood lymphocytes were analyzed in comparison with the isotype control mice (*n* = 10). The results showed that 60% (6/10) of the IL-10R blocker-treated mice still survived at 60 hpi, but all of the isotype group died at this time point (Figure 7A). Overall, IL-10R blocker-treated mice contribute to the enhanced survival of mice infected with FMDV. A further analysis showed that the number of peripheral blood lymphocytes in the IL-10R treatment group (16.8 ± 2.5 × 10^5^ cells/mL, *n* = 5, *p* = 0.0028) was significantly higher than that of the isotype group at 48 hpi (4.7 ± 1.3 × 10^5^ cells/mL, *n* = 5) (Figure 7B). An analysis of cell apoptosis demonstrated that, compared with mice of the isotype group, the level of apoptosis was significantly lower in T cells of mice treated by IL-10R at 48 hpi (T cells: 23.5 ± 3.0% vs. 16.2 ± 0.4%, respectively, *n* = 5, *p* = 0.0271) but not B and NK cells (B cells: 49.9 ± 3.3% vs. 41.0 ± 4.8%, *n* = 5, *p* = 0.4398, respectively, and NK cells: 39.8 ± 2.7% vs. 28.1 ± 2.0%, *n* = 5, *p* = 0.0772, respectively) (Figure 7C). To analyze if the IL-10R blocker could also inhibit the trafficking of lymphocytes from the peripheral blood to the heart, the frequencies of the CD3 T cells, CD19 B cells, and NK1.1 cells in the heart were analyzed by RT-qPCR at 48 hpi. The results showed that the CD19 mRNA levels significantly decreased in the heart of the IL-10R group when compared to the isotype group, and the CD3 and NK1.1 mRNA levels in the IL-10R group show no obvious differences from those that of the isotype group. It indicated that the IL-10R blocker can inhibit the trafficking of CD19^+^ B cells from the peripheral blood to the heart (Figure 7D). Similarly, the expression of CTLA-4 was significantly decreased in the CD4+ T cells of mice treated by IL-10R at 48 hpi compared with mice of the isotype group (3.6 ± 0.3% vs. 5.4 ± 0.5%, respectively, *n* = 5, *p* = 0.0430), and the expression of TIGIT and 2B4 was significantly decreased in CD8+ T cells of the mice treated by IL-10R at 48 hpi compared with mice of the isotype group (CD8-TIGIT: 1.6 ± 0.3% vs. 10.4 ± 3.1% and CD8-2B4: 1.2% vs. 18.8 ± 2.0%, *n* = 5, *p* = 0.0234 and *p* = 0.0010, respectively) (Figure 7E). Overall, these results indicated that the IL-10R blocker contributed to the enhance survival of mice infected with FMDV through alleviating lymphopenia.

### 3.9. Knocking Out IL-10 Prevented Lymphopenia Driven by FMDV Infection

To further determine the role of IL-10 in lymphopenia, IL-10^-/-^ mice was used to further observe the survival time of mice and the number of peripheral blood lymphocytes. The results showed that 77.8% (7/9) of IL-10^-/-^ mice still survived at 60 hpi, but all wild-type mice died at this time point (Figure 8A). A further analysis showed that the number of peripheral blood lymphocytes from IL-10^-/-^ mice (99.8 ± 14.5 × 10^4^ cells/mL, *n* = 5, *p* = 0.0003) was significantly higher than that of wild-type mice at 48 hpi (8.02 ± 2.8 × 10^4^ cells/mL, *n* = 5) (Figure 8B). An analysis of cell apoptosis demonstrated that, compared with wild-type mice, the level of apoptosis was significantly lower in the T cells and NK cells of IL-10^−/−^ mice at 48 hpi (T cells: 51.9± 4.4% vs. 34.9 ± 2.1%, respectively, *n* = 5, *p* = 0.0244 and NK cells: 41.8 ± 3.6% vs. 29.5 ± 2.2%, respectively, *n* = 5, *p* = 0.0447) but not B cells (49.1± 5.5% vs. 42.7 ± 3.4%, respectively, *n* = 5, *p* = 0.5371) (Figure 8C). Similarly, the CD19 mRNA levels significantly decreased in the hearts of IL-10^-/-^ mice when compared to the wild-type mice, but the NK1.1 mRNA levels significantly increased in the hearts of IL-10^-/-^ mice when compared to the wild-type mice, and the CD3 mRNA levels of IL-10^-/-^ mice showed no obvious differences with that of the wild-type mice. It further supported that IL-10 mediated the trafficking of CD19+ B cells from the peripheral blood to the heart (Figure 8D). The expression of TIGIT, CTLA-4, and PD-1 was significantly decreased in the CD4+ T cells of IL-10^-/-^ mice at 48 hpi compared with that of wild-type mice (CD4-TIGIT: 1.6 ± 0.2% vs. 3.5 ± 0.5%, CD4-CTLA-4: 1.8 ± 0.1% vs. 2.8 ± 0.4%, and CD4-PD-1: 4.9 ± 0.4% vs. 9.6 ± 0.5%, *n* = 5, *p* = 0.0056, *p* = 0.0360, and *p* = 0.0002, respectively), and the expression of TIGIT, PD-1, and 2B4 was significantly decreased in the CD8+ T cells of IL-10^-/-^ mice at 48 hpi compared with that of the wild-type mice (CD8-TIGIT: 1.7 ± 0.1% vs. 5.3 ± 0.6%, CD8-PD-1: 2.8 ± 0.4% vs. 12.9 ± 0.5%, and CD8-2B4: 1.8 ± 0.4% vs. 17.2 ± 3.3%, *n* = 5, *p* = 0.0004, *p* < 0.0001, and *p* = 0.0017, respectively) (Figure 8E). Overall, these results from the IL-10^-/-^ mice further indicated that IL-10 plays an important role in lymphopenia caused by FMDV infection.

## 4. Discussion

Viral infections leading to lymphopenia have been widely reported, such as severe acute respiratory syndrome coronavirus 2 (SARS-CoV-2) [32], Ebola virus (EBOV) [33], human immunodeficiency virus (HIV) [34], measles virus (MV) [35], influenza A virus (IAV) [36], and African swine fever virus (ASFV) [37]. Recent studies have indicated that lymphopenia was associated with the disease severity of patients, because individuals who died of SARS-CoV-2 and EBOV infection had expressively lower lymphocyte counts than survivors [38,39]. During FMDV infection, more profound lymphopenia was more prone to be induced by more virulent strains (O1 Campos, O Taiwan 97, A24 Cruzeiro, C3 Resende, and Cs8C1) than the less virulent strains (O SK 2000 and A12) [3,4,26]. In this study, lymphopenia was observed in C57BL/6 mice infected with the FMDV serotype O, and the mice with severe clinical symptoms had significantly lower lymphocyte counts than mild or without clinical symptoms (*p* < 0.001), which indicated that the level of lymphocyte counts was related to the disease severity during FMDV infection.

Lymphopenia is likely to delay viral clearance in favor of macrophage stimulation and the accompanying “cytokine storm”, which results in the dysfunction of host organs [40]; these damages could increase the risk of developing opportunistic infections. A report described that lymphopenia in COVID-19 patients could increase the risk of developing opportunistic infections of mucormycosis, while the recovery of the lymphocyte count could improve the acquired immune response and induce the production of mucoralean-specific T cells [41]. Similarly, opportunistic infection is a significant feature during HIV infection, which can result in high morbidity and even mortality [42,43]. To the best of our knowledge, there are no reports of opportunistic infections in FMD. It is interesting to further investigate whether opportunistic infections accompany lymphopenia caused by FMDV infection in the future.

A previous study reported that the mRNA expression of caspase 3 was upregulated in PBMC from infected pigs compared to uninfected controls, which supported that lymphocytes apoptosis may be involved in lymphopenia [3]. In this study, the apoptosis of the T cells, B cells, and NK cells in the peripheral blood was detected by Annexin V^+^/PI^+^ detection following FMDV infection. To investigate whether lymphocyte apoptosis is related to FMDV infection, the RdRp and VP1 sequences of FMDV were detected in T, B, and NK cells from mice infected with FMDV by flow sorting in our study. The results showed that no sequence of FMDV was detected by RT-qPCR, which supported that these lymphocytes could not be infected with FMDV serotype O [3,9]. Thus, it was believed that lymphocyte apoptosis was caused by indirect factors, such as the activation of apoptosis related-receptors (Fas/FasL) [36], the interaction between the viral protein and host cellular receptors [44], and cytokines (TNF-α) [45]. Interestingly, IL-10 but not TNF-α in the serum from infected mice was upregulated following FMDV infection, and the elevated levels of IL-10 coincided with the occurrence of lymphopenia in this study. IL-10, as an autocrine growth factor in malignant B-1 cells, lies in its ability to inhibit the apoptosis induction of B cells [46,47], which supported that anti-IL-10R antibody treatment or the knock out of IL-10 may not lower the level of apoptosis of B cells. However, IL -10 produced by CD9^+^ regulatory B cells of severe asthmatic patients induced T-cell apoptosis by the extrinsic and intrinsic apoptosis pathways via a MAPK-dependent mechanism [21]. Moreover, IL-10 overproduction in lupus induced lymphocyte apoptosis via the activation of Fas and caspase [48]. In this study, knocking out IL-10 and blocking IL-10/IL-10R signaling in vivo were able to inhibit the apoptosis of T lymphocytes. Moreover, IL-10 could induce T apoptosis in vitro (Appendix A), which was consistent with a previous report [21]. Thus, the elevated IL-10 in the serum is able to induce T-cell apoptosis in the peripheral blood, which contributes to lymphopenia during FMDV infection. It is interesting to further the molecular mechanism of IL-10 induing T-lymphocyte apoptosis during FMDV infection.

Type 1 IFN was originally identified as a humoral factor that confer an antiviral state on cells [49], but it was also found to regulate lymphocyte recirculation and cause transient lymphopenia [10]. The elevated IFN-α level was observed in cattle and pigs infected with FMDV, and it was speculated that the tissue redistribution of lymphocytes may be involved in lymphopenia [11]. In this study, lymphocytes migrated from the peripheral blood to the heart after FMDV infection by tracking CFSE-labeled lymphocytes. Interestingly, the levels of IFN-α/β reported to be involved in trafficking were not upregulated. In addition, the levels of IL-6, IL-2, IL-17, IL-18, IL-1β, TNF-α, IFN-α/β, TGF-β, and CXCL1 in the serum of infected mice were similar with that from mock mice. Notably, elevated IL-10 not only coincided with the occurrence of lymphopenia but, also, with elevated levels of CXCL13. Interestingly, knocking out IL-10 and blocking IL-10/IL-10R signaling in vivo could inhibit the trafficking of CD19+ B cells from the peripheral blood to the heart. These results supported that lymphocyte trafficking may be related to IL-10 or CXCL13 rather than type I IFNs. CXCL13, also known as B cell-attracting chemokine-1, could attract B cells to secondary lymphoid tissue. It is expressed by follicular dendritic cells (FDCs), stromal cells, monocytes, and macrophages [50,51,52]. A recent study reported that IL-10 can control the expression of CXCL13 via the JAK/STAT pathway [53]. Therefore, it was speculated that IL-10 may mediate B cell trafficking by regulating CXCL13 expression.

T-cell exhaustion is a state of T-cell dysfunction that arises during many chronic infections and cancers. It is defined by a poor effector function, the sustained expression of inhibitory receptors, and a transcriptional state distinct from that of a functional effector or memory T cells [54,55]. Previous studies have reported that lymphopenia was accompanied by the upregulated expression of coinhibitory molecules during viral infections, including SARS-CoV-2- [56,57], HIV [58], EBOV [59], and IAV infection [60]. Notably, the expression of PD-1, LAG-3, and TIGIT in CD4+ T cells showed the strongest inverse associations with CD4+ T-cell counts during HIV infection [58], and the blockade of PD-L1 in vivo improves the number of CD8^+^ T cells in high pathological IAV infection [60]. The downstream effects of PD-1 signaling include the inhibition of AKT, phosphoinositide 3-kinase, extracellular signal-regulated kinase, and phosphoinositide phospholipase C-γ and regulation of the cell cycle, leading to decreased IFN-γ/IL-2 production, reduced proliferation, and increased risk for apoptosis [61]. In this study, the upregulated expression of CTLA-4, LAG-3, 2B4, and TIGIT were observed in CD8^+^ T cells and CD4^+^ T cells of mice infected with FMDV, which would inhibit T-cell proliferation and promote T-cell apoptosis during virus infections [56,57]. As per an earlier report, IL-10 could induce PD-L1 expression on peripheral blood monocytes via STAT3 [62]. In this study, knocking out IL-10 and blocking IL-10/IL-10R signaling in vivo downregulated the expression of TIGIT and 2B4 on CD8^+^ T cells and CTLA-4 on CD4^+^ T cells. Thus, IL-10^-/-^ and blocking IL-10/IL-10R signaling contribute to T-cell proliferation and inhibit T-cell apoptosis via regulating the expression of coinhibitory molecules on T cells.

## 5. Conclusions

Lymphopenia involving in CD4^+^ and CD8^+^ T cells, B cells, and NK cells was observed in C57BL/6 mice infected with FMDV. Lymphopenia was associated with disease severity. Elevated IL-10 in the serum from infected mice was related to the occurrence of lymphopenia, and the knocking out of IL-10 and blocking IL-10/IL-10R signaling prevent lymphopenia via inhibiting apoptosis, trafficking, and the expression of coinhibitory molecules of lymphocytes, which contributes to enhancing the survival of mice infected with FMDV. Our results indicate that knocking out IL-10 and blocking IL-10/IL-10R signaling may contribute to preempting disease progression in FMDV-infected animals with low lymphocyte counts, which could be considered as a treatment strategy for FMD animals with high IL-10 levels.

## Figures and Tables

**Figure 1 viruses-13-02358-f001:**
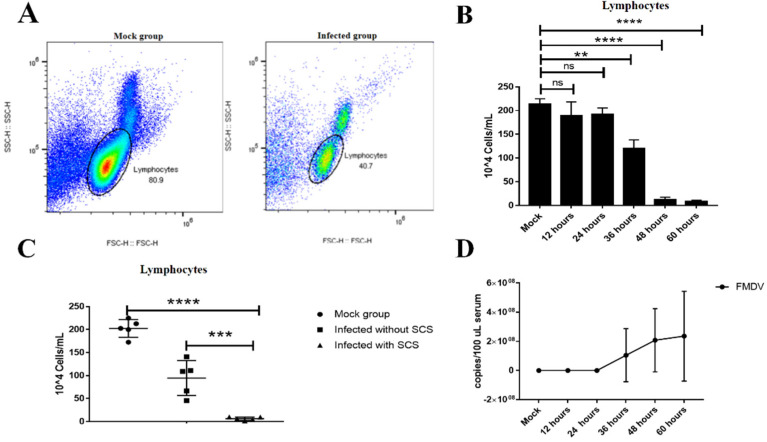
Lymphopenia coincided with FMDV RNA in the serum during FMDV infection. (**A**) Representative FACS plots of lymphocytes from mock mice and infected mice at 48 hpi. (**B**) The number of lymphocytes in the peripheral blood were shown in mock mice and infected mice at 12, 24, 36, 48, and 60 hpi. (**C**). The number of lymphocytes in the peripheral blood were shown in mock mice, infected mice with severity clinical symptoms (SCS), and infected mice without SCS (including mild clinical symptoms). (**D**) FMDV RNA copies in the serum from mock mice and infected mice at 12, 24, 36, 48, and 60 hpi were determined by RT-qPCR. The significance was detected using an unpaired *t*-test. **, *p*  ≤ 0.01; ***, *p*  ≤ 0.001; ****, *p*  ≤ 0.0001. ns indicates a non-statistically significant difference (*p* > 0.05).

**Figure 2 viruses-13-02358-f002:**
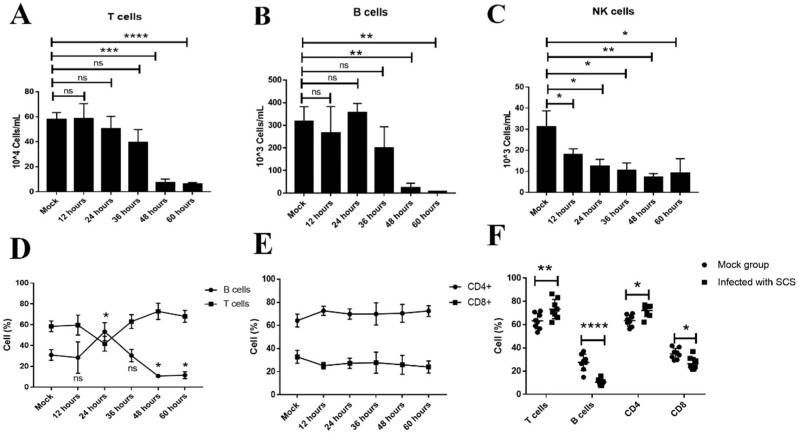
The numbers and frequencies of T cells, B cells, and NK cells in the peripheral blood were decreased in FMDV-infected mice. The numbers of T cell (**A**), B cells (**B**), and NK cells (**C**) in the peripheral blood from mock mice and FMDV-infected mice at 12, 24, 36, 48, and 60 hpi (**D**). The frequencies of the T cells and B cells were measured from mock mice and FMDV-infected mice at 12, 24, 36, 48, and 60 hpi (**E**). The frequencies of the CD4^+^ T cells and CD8^+^ T cells were measured from mock mice and FMDV-infected-mice at 12, 24, 36, 48, and 60 hpi (**F**). The frequencies of the T cells, B cells, CD4^+^ T cells, and CD8^+^ T cells from mock mice and infected mice with SCS were measured during FMDV infection. The T, B, and NK cell counts were analyzed by an unpaired *t*-test. The percentages of T, B, CD4+, and CD8+ T cells were analyzed by two-way ANOVA with Tukey’s multiple comparison test. *, *p* ≤ 0.05; **, *p* ≤ 0.01; ***, *p* ≤ 0.001; ****, *p* ≤ 0.0001. ns indicates a non-statistically significant difference (*p* > 0.05).

**Figure 3 viruses-13-02358-f003:**
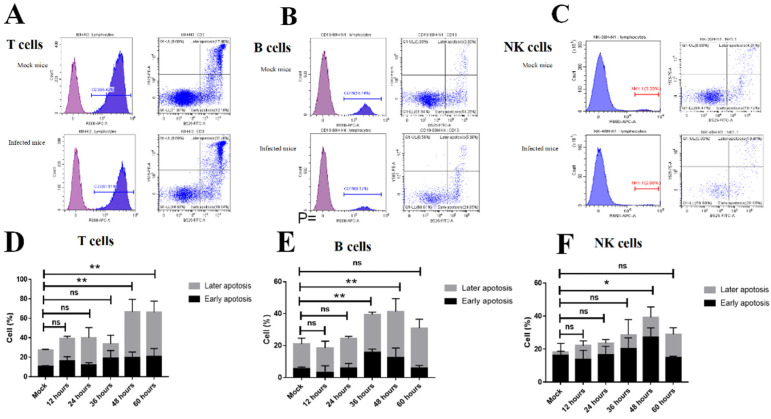
The apoptosis was detected in T cells, B cells, and NK cells in the peripheral blood from mock mice and FMDV-infected mice. Representative FACS plots of apoptosis in T cells (**A**), B cells (**B**), and NK cells (**C**) from mock mice and FMDV-infected mice. The apoptosis was detected in T cells (**D**), B cells (**E**), and NK cells (**F**) from mock mice and FMDV-infected mice at 12, 24, 36, 48, and 60 hpi by Annexin V/PI detection. Annexin V^+^/PI^−^ cells were considered early apoptotic cells, and Annexin V^+^/PI^+^ cells were considered late apoptotic cells. The apoptosis of the T, B, and NK cells was analyzed by two-way ANOVA with Tukey’s multiple comparison test. Significance was detected using an unpaired *t*-test. *, *p* ≤ 0.05; **, *p  *≤ 0.01. ns indicates a non-statistically significant difference (*p* > 0.05).

**Figure 4 viruses-13-02358-f004:**
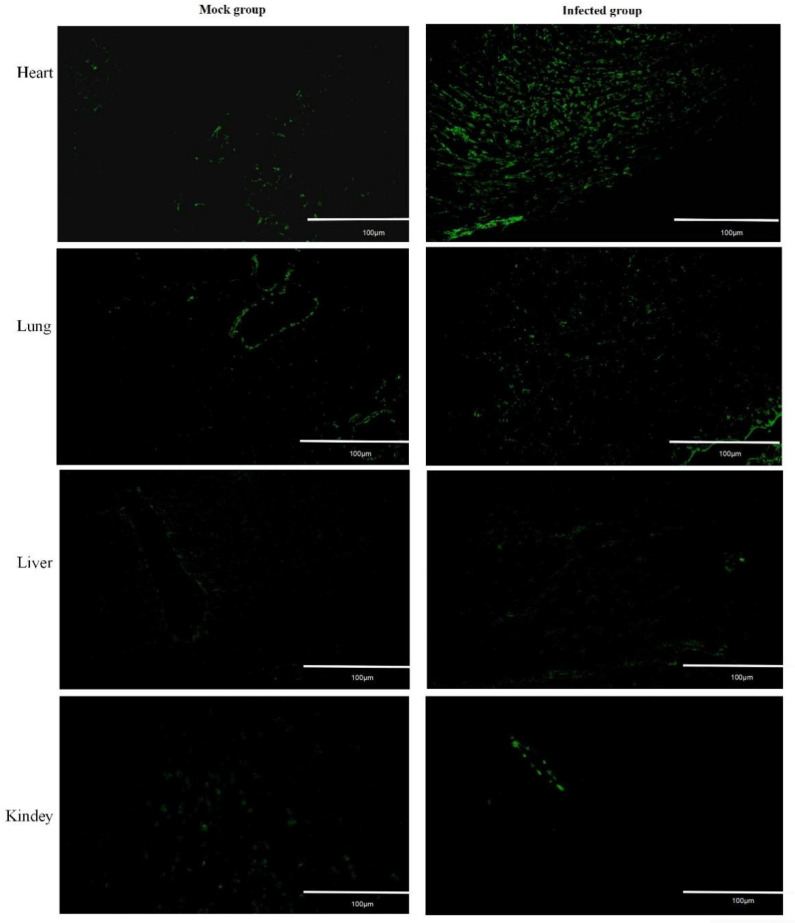
The CFSE-labeled lymphocytes migrate from the blood to tissue. The CFSE-labeled lymphocytes in the heart, liver, kidney, and lung from mock mice and infected mice were observed by fluorescence microscope (10×).

**Figure 5 viruses-13-02358-f005:**
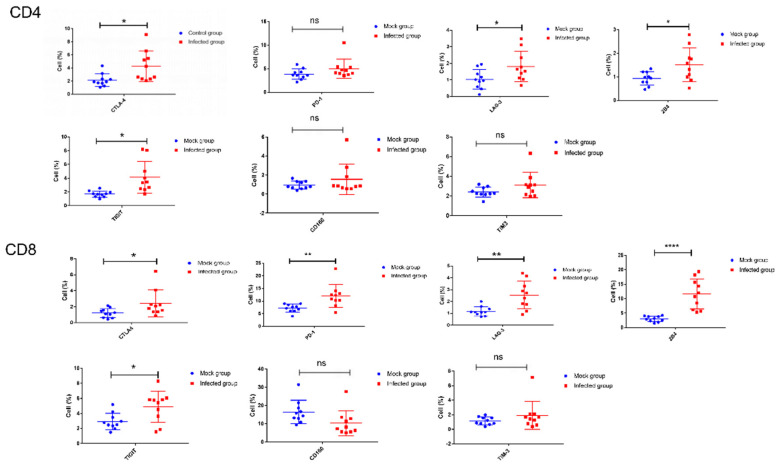
The expression of PD-1, CTLA-4, Tim-3, LAG-3, 2B4, TIGIT, and CD160 was analyzed on CD4 and CD8 T cells from mock mice and FMDV-infected mice. The expression of all known seven coinhibitory molecules was analyzed on peripheral blood T lymphocytes, which were harvested from the mock mice and FMDV-infected mice at 48 hpi (*n*  =  10 per group). Significance was detected using an unpaired *t*-test. *, *p* ≤ 0.05; **, *p* ≤ 0.01; ****, *p* ≤ 0.0001. ns indicates a non-statistically significant difference (*p* > 0.05).

**Figure 6 viruses-13-02358-f006:**
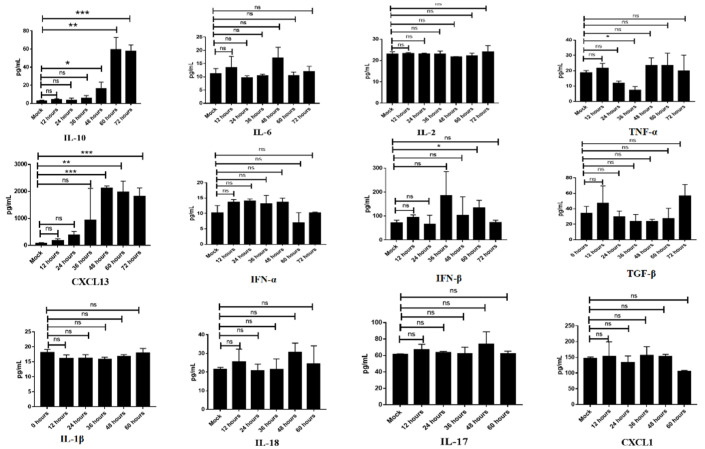
Potential cytokines/chemokines were detected in the serum from mock mice and FMDV-infected mice. The circulating levels (mean ± SEM) of various cytokines were measured by ELISA in the serum of mock mice and FMDV-infected mice at 12, 24, 36, 48, 60, and 72 hpi. The results are representative of a single experiment with 3 mice per time point. Significance was detected using an unpaired *t*-test. *, *p* ≤ 0.05; **, *p*  ≤ 0.01; ***, *p* ≤ 0.001. ns indicates a non-statistically significant difference (*p* > 0.05).

**Figure 7 viruses-13-02358-f007:**
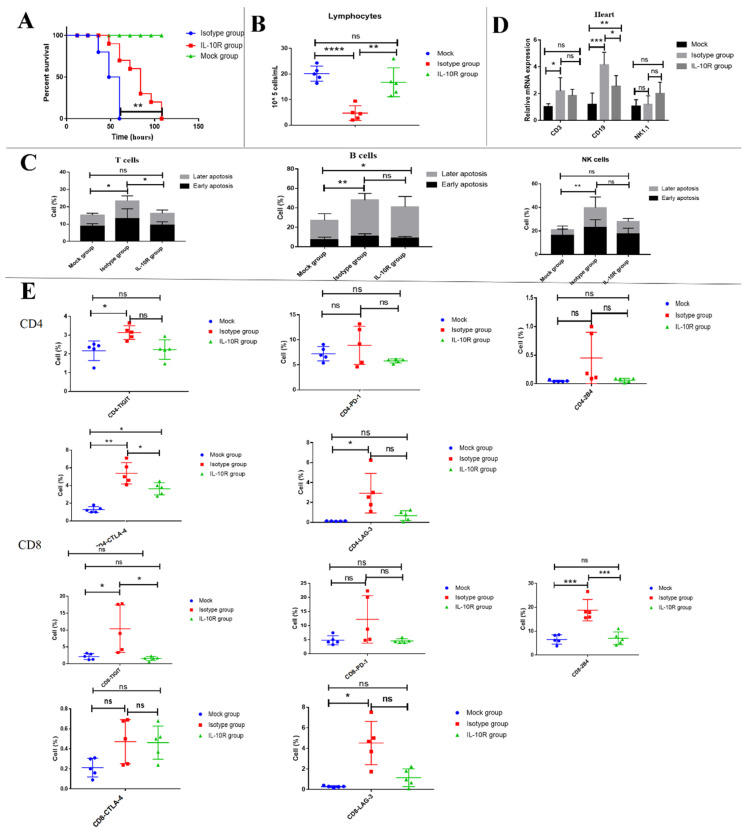
Blocking IL-10/IL-10R signaling with the IL-10R blocker prevented lymphopenia driven by FMDV infection. (**A**) The survival curve of mice treated with in vivo anti-mouse IL-10R (*n* = 10) and mice treated with in vivo rat IgG1 isotype control (*n* = 10) after FMDV infection. (**B**) The number of lymphocytes in the peripheral blood from mice treated with in vivo rat IgG1 isotype control (*n* = 5) and mice treated with in vivo anti-mouse IL-10R (*n* = 5). (**C**) The apoptosis of T cells and B cells in the peripheral blood from mice treated with in vivo rat IgG1 isotype control (*n* = 5) and mice treated with in vivo anti-mouse IL-10R (*n* = 5). (**D**) The frequencies of CD3, CD19, and NK1.1 cells in the heart from mice treated with in vivo rat IgG1 isotype control (*n* = 3) and mice treated with in vivo anti-mouse IL-10R (*n* = 3) are shown. (**E**) The expression of TIGIT, PD-1, 2B4, CTLA-4, and LAG-3 on CD4 and CD8 in the peripheral blood from mice treated with in vivo rat IgG1 isotype control (*n* = 5) and mice treated with in vivo anti-mouse IL-10R (*n* = 5). The survival rates in mice were compared by the log-rank test. The lymphocytes counts and the expression of coinhibitory molecules were analyzed by an unpaired *t*-test. The apoptosis of T, B, and NK cells and the mRNA expression of T, B, and NK cells in the heart were analyzed by two-way ANOVA with Tukey’s multiple comparison test. *, *p* ≤ 0.05; **, *p*  ≤ 0.01; ***, *p* ≤ 0.001; ****, *p* ≤ 0.0001. ns indicates a non-statistically significant difference (*p* > 0.05).

**Figure 8 viruses-13-02358-f008:**
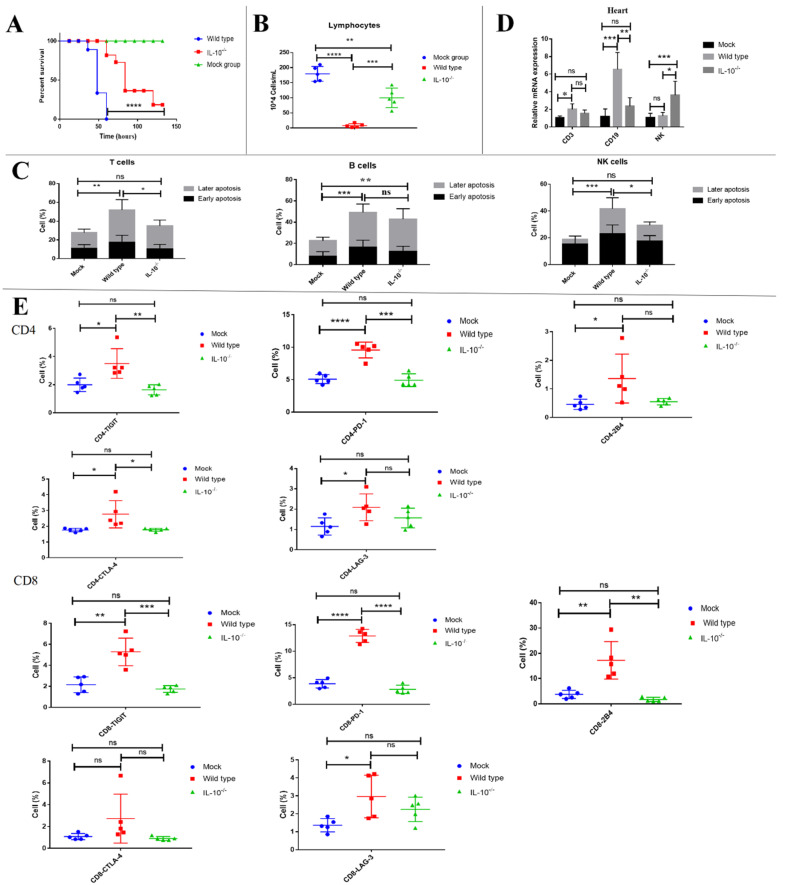
Knocking out IL-10 prevented lymphopenia driven by FMDV infection. (**A**) The survival curve of IL-10^-/-^ mice (*n* = 9) and wild-type mice (*n* = 9) infected with FMDV. (**B**) The number of lymphocytes in the peripheral blood from IL-10^-/-^ mice (*n* = 5) and wild-type mice (*n* = 5) at 48 hpi. (**C**) The apoptosis of T cells, B cells, and NK cells in the peripheral blood from IL-10^-/-^ mice (*n* = 5) and wild-type mice (*n* = 5) at 48 hpi. (**D**) The frequency of CD3, CD19, and NK1.1 cells in the heart from IL-10^-/-^ mice (*n* = 5) and wild-type mice (*n* = 5) at 48 hpi. (**E**) The expression of TIGIT, PD-1, 2B4, CTLA-4, and LAG-3 on CD4 and CD8 in the peripheral blood of IL-10^-/-^ mice (*n* = 5) and wild-type mice (*n* = 5) at 48 hpi. The survival rates in mice were compared by the log-rank test. The lymphocyte counts and the expression of coinhibitory molecules were analyzed by an unpaired *t*-test. The apoptosis of T, B, and NK cells and the mRNA expression of T, B, and NK cells in the heart were analyzed by two-way ANOVA with Tukey’s multiple comparison test. *, *p* ≤0.05; **, *p*  ≤ 0.01; ***, *p* ≤ 0.001; ****, *p* ≤ 0.0001. ns indicates a non-statistically significant difference (*p* > 0.05).

## Data Availability

Not applicable.

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
