# Peer review of "Interleukin-10-Mediated Lymphopenia Caused by Acute Infection with Foot-and-Mouth Disease Virus in Mice"

_viruses, 2021, doi:10.3390/v13122358_

Round 1

Reviewer 1 Report

The manuscript describes the role of Interleukin-10 in mediating lymphopenia after acute infection with FMD virus in mice.

Title: The title has to reflect the species that was used in the study.  Consider changing the title as “Interleukin-10 mediated lymphopenia caused by acute infection with foot-and-mouth disease virus in mice” .

Since the manuscript has too many acronyms, expansions must be provided at the beginning of the manuscript.

Introduction:

Line 30: Change from ‘contagious’ to ‘infectious’.

Line 38-39: Change to: ‘Following FMDV infection, lymphocyte depletion in peripheral blood, referred to as lymphopenia, is a common characteristic in pigs [3,4], cattle [5,6] and C57BL/6 mice [7]’.

Line 42: Start as ‘A previous study’

Lines 44-48: Sentence confusing.  Please reframe the sentence as ‘Despite high viremia in pigs infected with FMDV (serotypes of C, O, and A), no viral RNA or virus could be isolated on BHK-21 cells from the lysate of the peripheral blood mononuclear cells (PBMCs) indicating that active infection of lymphocytes is not believed to be responsible for the lymphopenia [3,9].’

Line 49-50: Are you referring to Type 1 interferons? If yes, correct as ‘Type 1 interferons (Type 1 IFN).  What sub-species of IFN must be mentioned (alpha, beta or gamma).

Line 55: Change to ‘IL-10, a master regulator of anti-inflammatory response, is critical to protect the host from tissue damage during acute phases of immune responses [12].

Line 57-60: Change to ‘Despite its primary anti-inflammatory role, IL-10 plays a dual role of promoting pathogen persistence [14-17] and limiting immune pathology [18,19] and these effects may be due to the IL-10 producing immune cell, the available IL-10 levels, the target cell, and the microenvironmental co-factors, such as other cytokines [20].’

Line 61: Start as ‘It has been suggested that ….’

Materials and Methods:

Line 86-90: All infectious animal work were performed in the biosafety level 3 biocontainment laboratory at LVRI, in strict accordance with good animal practice as stipulated by the Animal Ethics Procedures and Guidelines of the People's Republic of China, and the study was approved by the Animal Ethics Committee of LVRI, CAAS (furnish the application number and date of approval).

Line 95-97: Provide a reference to the statement.  Change to ‘hours post-inoculation’

Line 98-99: Furnish the time points of sample collection here.

Line 100: Provide a reference to the statement.

Line 106-110: Change as follows: ‘Anti-mouse CD3, CD19, NK1.1, CD8, and CD4 were used to label T lymphocytes, B lymphocytes, NK lymphocytes, CD3+CD8+ T lymphocytes and CD3+CD4+ T lymphocytes respectively. Anti-mouse PD-1, CTLA-4, Tim-3, LAG-3, 2B4, TIGIT, CD160, and isotype control of these antibodies were used to evaluate CD4+ and 110

CD8+ T cells exhaustion.’

Line 122: Replace the ‘full-stop’ with a ‘comma’ after the reference [27], and delete ‘Injections wre given on…’ to merge the two sentences.

Line 126: Change from ‘stained’ to ‘labelled’.

Line 128: delete the word ‘cells’.

Line 131: Replace the ‘full-stop’ with a ‘comma’ after the reference [28], and delete ‘Apoptosis analysis was detected by’ and replace with ‘using’ to merge the two sentences.

Line 134: ‘Viraemia’ is a condition following virus infection.  But you are detecting viraemia using ‘viral RNA’ as a measure.  Therefore, replace the word ‘Viraemia’ with ‘FMDV RNA’.  Please check if the reference [29] is appropriate here.

Line 139: There is no reference to mock-mice in section 2.2. Please add and then change as ‘mock- or infected-mice’ throughout the manuscript.

Line 142-147: Change to ‘RT-qPCR’.  No results are presented in the results section for this work.

Line 151-172: Merge Sections 2.7 and 2.8 and provide an appropriate title.  Currently the two sections look disjointed and not related to each other.

Results:

In this section, please provide p-values when you are mentioning statistical significance between groups.  Please use viral RNA load or viral RNA copies when you are refereeing to viral loads.  As per your methods, you are not titrating the virus but quantifying the FMDV RNA.

Line 186-187.  Please describe the expected symptoms of FMD in mice in the methods section 2.2.

Line 187: Should it be ‘lethargy’ instead of ‘quietness’?

Line 197: Delete the words in the brackets.  This has been defined in the methods section 2.2.

Line 204: Figure 1: Change to ‘FMDV RNA copies’

Line 206: Change title to ‘Lymphopenia in mice acutely infected with FMDV’

Line 213: Furnish the p-value.

Line 219: Change to ‘viral RNA loads’.

Line 223: Delete the sentence ‘Lymphocytes are consisted of T cells, B cells and NK cells’.

Discussion:

Wherever you refer to a previous study, please start as ‘A previous study reported’ or ‘As per an earlier report/study’ or According to a previous study/report’.

Line 80: Change to ‘Type 1 IFN’

Conclusions:

Line 520-521: Change to ‘CD4+ and CD8+ T cells’, delete the word ‘obviously’ and end the sentence at ‘FMDV’.  Start a new sentence from ‘Lymphopenia was….’

Line 522: Delete ‘Importantly,’ and start as ‘Elevated IL-10…’

Line 526: Delete ‘These results indicated’ and change to ‘Our results indicate’

Author Response

Title: The title has to reflect the species that was used in the study. Consider changing the title as “Interleukin-10 mediated lymphopenia caused by acute infection with foot-and-mouth disease virus in mice”.

Response: Thank you for your suggestion. The title has been revised in the revised version of the manuscript.

Since the manuscript has too many acronyms, expansions must be provided at the beginning of the manuscript.

Response: Thank you for your suggestion. The abbreviation words have been listed in lines 29-37 of the revised version of the manuscript.

Introduction:

Line 30: Change from ‘contagious’ to ‘infectious’.

Response: It has been revised in the revised version of the manuscript.

Line 38-39: Change to: ‘Following FMDV infection, lymphocyte depletion in peripheral blood, referred to as lymphopenia, is a common characteristic in pigs [3,4], cattle [5,6] and C57BL/6 mice [7]’.

Response: It has been revised in lines 50-51 of the revised version of the manuscript.

Line 42: Start as ‘A previous study’.

Response: It has been revised in the revised version of the manuscript.

Lines 44-48: Sentence confusing. Please reframe the sentence as ‘Despite high viremia in pigs infected with FMDV (serotypes of C, O, and A), no viral RNA or virus could be isolated on BHK-21 cells from the lysate of the peripheral blood mononuclear cells (PBMCs) indicating that active infection of lymphocytes is not believed to be responsible for the lymphopenia [3,9].’

Response: Thank you for your kind suggestion. The sentence has been revised in lines 56-60 of the revised version of the manuscript.

Line 49-50: Are you referring to Type 1 interferons? If yes, correct as ‘Type 1 interferons (Type 1 IFN).  What sub-species of IFN must be mentioned (alpha, beta or gamma).

Response: The sentence has been revised as “IFN-α/β of Type 1 interferons (Type 1 IFN) reported to be related to trafficking of lymphocytes was increased following FMDV infection ...” in the lines 61-62 of the revised version of the manuscript.

Line 55: Change to ‘IL-10, a master regulator of anti-inflammatory response, is critical to protect the host from tissue damage during acute phases of immune responses [12].

Response: Thank you for your suggestion. The sentence has been revised in lines 67-68 of the revised version of the manuscript.

Line 57-60: Change to ‘Despite its primary anti-inflammatory role, IL-10 plays a dual role of promoting pathogen persistence [14-17] and limiting immune pathology [18,19] and these effects may be due to the IL-10 producing immune cell, the available IL-10 levels, the target cell, and the microenvironmental co-factors, such as other cytokines [20].’

Response: Thank you for your suggestion. The sentence has been revised in lines 70-74 of the revised version of the manuscript.

Line 61: Start as ‘It has been suggested that ….’

Response: Thank you for your suggestion. The sentence has been revised in line 73-74 of the revised version of the manuscript.

Materials and Methods:

Line 86-90: All infectious animal work were performed in the biosafety level 3 biocontainment laboratory at LVRI, in strict accordance with good animal practice as stipulated by the Animal Ethics Procedures and Guidelines of the People's Republic of China, and the study was approved by the Animal Ethics Committee of LVRI, CAAS (furnish the application number and date of approval).

Response: Thank you for your suggestion. The approval code and approval data have been added, and the sentence has been revised in lines 98-132 of the revised version of the manuscript.

Line 95-97: Provide a reference to the statement. Change to ‘hours post-inoculation’

Response: A previous study showed that Swiss, BALB/ C and C57BL/6 mice were susceptible to infection with 105 PFUs of FMDV C-S8c1 and showed symptoms or death following viral infection (Salguero et al., 2005). Based on this result, our laboratory found that 5×105.5 TCID50 per mice was the lowest dose of infection to cause death in C57BL/6 mice at age of 4-6 weeks. In addition, “hours postinoculation” has been revised as “hours post-inoculation” in the revised version of the manuscript.

Line 98-99: Furnish the time points of sample collection here.

Response: Thank you for your suggestion. The time points of serum sample collection have been added in line 146 of the revised version of the manuscript.

Line 100: Provide a reference to the statement.

Response: We are apologized for this inappropriate description. To avoid confusion, the sentence has been moved to 2.6 and revised to “FMDV RNA in serums was quantitatively determined by RT-qPCR and the primers targeting FMDV 3D gene were reported in a previous study [30]” in the revised version of the manuscript.

Line 106-110: Change as follows: ‘Anti-mouse CD3, CD19, NK1.1, CD8, and CD4 were used to label T lymphocytes, B lymphocytes, NK lymphocytes, CD3+CD8+ T lymphocytes and CD3+CD4+ T lymphocytes respectively. Anti-mouse PD-1, CTLA-4, Tim-3, LAG-3, 2B4, TIGIT, CD160, and isotype control of these antibodies were used to evaluate CD4+ and CD8+ T cells exhaustion.’

Response: Thank you for your suggestion. The sentence has been revised in lines 154-158 of the revised version of the manuscript.

Line 122: Replace the ‘full-stop’ with a ‘comma’ after the reference [27], and delete ‘Injections were given on…’ to merge the two sentences.

Response: Thank you for your suggestion. The sentence has been revised lines 167-170 of the revised version of the manuscript.

Line 126: Change from ‘stained’ to ‘labelled’.

Response: It has been revised in the revised version of the manuscript.

Line 128: delete the word ‘cells’.

Response: It has been deleted in the revised version of the manuscript.

Line 131: Replace the ‘full-stop’ with a ‘comma’ after the reference [28], and delete ‘Apoptosis analysis was detected by’ and replace with ‘using’ to merge the two sentences.

Response: The sentence has been revised in line 178 of the revised version of the manuscript.

Line 134: ‘Viraemia’ is a condition following virus infection.  But you are detecting viraemia using ‘viral RNA’ as a measure. Therefore, replace the word ‘Viraemia’ with ‘FMDV RNA’.  Please check if the reference [29] is appropriate here.

Response: Thank you for your suggestion. The sentence has been revised as “FMDV RNA in serums was quantitatively determined by RT-qPCR, and the primers targeting FMDV 3D gene have been reported in a previous study [30].” in lines 235-236 of the revised version of the manuscript.

Line 139: There is no reference to mock-mice in section 2.2. Please add and then change as ‘mock- or infected-mice’ throughout the manuscript.

Response: Thank you for your suggestion. Mock-mice were inoculated subcutaneously with 0.05 mL phosphate buffered saline (PBS). The description has been added in lines 141-142, and “mock- or infected-mice”has been revised throughout the revised version of the manuscript. 

Line 142-147: Change to ‘RT-qPCR’.  No results are presented in the results section for this work.

Response: Thank you for your suggestion. “RT-qPCR” has been revised throughout the manuscript. In addition, the results on the mRNA expression of CD3 T cells, CD19 B cells, and NK1.1 cells in heart has been shown in lines 648-652 and 700-705 of the revised version of the manuscript.

Line 151-172: Merge Sections 2.7 and 2.8 and provide an appropriate title. Currently the two sections look disjointed and not related to each other.

Response: Thank you for your suggestion. Sections 2.7 and 2.8 in the previous manuscript has been merged into 2.7 section titled “Tracking lymphocytes labeled with CFSE in vivo” in the revised version of the manuscript.

Results:

In this section, please provide p-values when you are mentioning statistical significance between groups.  Please use viral RNA load or viral RNA copies when you are refereeing to viral loads. As per your methods, you are not titrating the virus but quantifying the FMDV RNA.

Response: Thank you for your suggestion. P-values mentioning statistical significance between groups has been added in throughout revised manuscript. In addition, “virus loads” in the previous manuscript has been revised as “viral RNA loads” in lines 383 and 386 of the revised version of the manuscript.

Line 186-187. Please describe the expected symptoms of FMD in mice in the methods section 2.2.

Response: The expected symptoms of mice infected with FMDV has been added in lines 142-143 of section 2.2 in the revised version of the manuscript.

Line 187: Should it be ‘lethargy’ instead of ‘quietness’?

Response: It has been revised in line 309 of the revised version of the manuscript.

Line 197: Delete the words in the brackets. This has been defined in the methods section 2.2.

Response: It has been deleted in the revised version of the manuscript.

Line 204: Figure 1: Change to ‘FMDV RNA copies’

Response: It has been revised in line 326 of the revised version of the manuscript.

Line 206: Change title to ‘Lymphopenia in mice acutely infected with FMDV’

Response: It has been revised in line 330 of the revised version of the manuscript.

Line 213: Furnish the p-value.

Response: It has been added in lines 334-380 of the revised version of the manuscript.

Line 219: Change to ‘viral RNA loads’.

Response: It has been revised in lines 382-383 of the revised version of the manuscript.

Line 223: Delete the sentence ‘Lymphocytes are consisted of T cells, B cells and NK cells’.

Response: The sentence has been deleted in the revised version of the manuscript.

Discussion:

Wherever you refer to a previous study, please start as ‘A previous study reported’ or ‘As per an earlier report/study’ or According to a previous study/report’.

Response: It has been revised throughout the revised version of the manuscript.

Line 80: Change to ‘Type 1 IFN’

Response: It has been revised in line 776 of the revised version of the manuscript.

Conclusions:

Line 520-521: Change to ‘CD4+ and CD8+ T cells’, delete the word ‘obviously’ and end the sentence at ‘FMDV’.  Start a new sentence from ‘Lymphopenia was….’

Response: Thank you for your suggestion. The sentence has been revised in line 820-821 of the revised version of the manuscript.

Line 522: Delete ‘Importantly,’ and start as ‘Elevated IL-10…’

Response: It has been revised in line 822 of the revised version of the manuscript.

Line 526: Delete ‘These results indicated’ and change to ‘Our results indicate’

Response: It has been revised in lines 825-826 of the revised version of the manuscript.

Reviewer 2 Report

This manuscript by Guo et al. investigated the role of IL-10 in lymphopenia caused by foot-and-mouth disease virus infection. The study's major finding is that FMD virus-induced lymphopenia is mediated by the production of IL-10, maybe due to the induction of apoptosis and trafficking of lymphocytes in peripheral blood. In addition, blocking of IL-10 biological activities improves the survival of mice infected with FMD viruses, possibly leading to a novel therapeutic approach for FMD. This manuscript is well-written and organized. The results are very scientific and transparent.

Additionally, this manuscript contains valuable information for readers. However, the authors need to address some concerns, which need to be clarified before publication in Vaccines. Please see my comments, explained below.

Major comments:

  1. Abstract: The authors state IL-10-/- may represent a novel therapeutic approach, but this is impossible. Please consider modifying like “our finding support that blocking IL-10/IL-10R signaling may represent a novel therapeutic approach for FMD.”.
  2. Pages 1-2, lines 45-47: I did not understand the meaning of this sentence. Please re-write this. In addition, what are BHK-21 and its relevance to this study?
  3. Section 2.1.: The authors should state the experimental endpoint for in vivo infection study.
  4. Materials and methods section: The method of sacrifice should be addressed.
  5. Section 2.10.: Which method for statistical analysis should be stated more precisely.
  6. Figure captions for 1, 2, 3, 5, 6, 7, and 8: more specific methods of statistics should be mentioned.
  7. Figure 2D: Statistical analysis is mandatory.
  8. In figure 6, the authors showed IL-6 level was increased after 48 hours post-infection. In addition, in figure 2, decreased lymphocytes number were observed after 48 hours post-infection. If IL-10 is involved in the decreased lymphocytes number, the time gap among them can be observed.
  9. Figure 7C: Why anti-IL-10R antibody treatment did not lower the level of apoptosis of B cells.
  10. Figures 7 and 8: The statistics between mock and IL-10R groups are missing.
  11. Do you have any idea how the FMD virus induces IL-10 production?

Minor comments:

  1. “+” should be superscript, e.g., CD4+. Please carefully go through the manuscript and correct all of them.
  2. IFN, IL, MHC, EDTA, NK, hpi, TNF, TGF, PBS, and ELISA should be spelled out at their first appearance. Please carefully go through the manuscript and correct all of them.
  3. Page 7, line 294: Please correct “live” to “liver”.

Author Response

Major comments:

Abstract: The authors state IL-10-/- may represent a novel therapeutic approach, but this is impossible. Please consider modifying like “our finding support that blocking IL-10/IL-10R signaling may represent a novel therapeutic approach for FMD.”.

Response: Thank you for your suggestion. The sentence has been revised in lines 23-25 of the revised version of the manuscript.

Pages 1-2, lines 45-47: I did not understand the meaning of this sentence. Please re-write this. In addition, what are BHK-21 and its relevance to this study?

Response: The sentence has been re-written as “Despite high viremia in pigs infected with FMDV (serotypes of C, O, and A), no viral RNA or virus could be isolated on BHK-21 cells from the lysate of the peripheral blood mono-nuclear cells (PBMCs) indicating that active infection of lymphocytes is not believed to be responsible for the lymphopenia [3,9].” in the revised version of the manuscript according to Reviewer 1’s suggestion. BHK-21 cells are Baby Hamster Syrian Kidney cells, which are the cell line of choice for the isolation and propagation of FMDV as well as vaccine production worldwide (Dill et al., 2020). A previous study reported that FMDV could not be isolated on BHK-21 cells from the lysate of the peripheral blood mono-nuclear cells (PBMCs) from FMDV-infected pig with high viremia, which indicated that lymphocytes could not be infected with FMDV. Thus, active infection of lymphocytes is not believed to be responsible for lymphopenia [3,9].

Section 2.1.: The authors should state the experimental endpoint for in vivo infection study.

Response: Thank you for your suggestion. Animals were euthanized when infected mice showed severe respiratory distress, inability to maintain body temperature or secondary infection. The related description has been added in lines 132-134 of the revised version of the manuscript.

Materials and methods section: The method of sacrifice should be addressed.

Response: Before samples collection, mice were sacrificed using intraperitoneal (i.p.) injections with 200 μl pentobarbital, followed by cervical dislocation or exsanguination under terminal anesthesia. The relevant descriptions have been added in lines 150-152 of the revised version of the manuscript.

Section 2.10.: Which method for statistical analysis should be stated more precisely.

Response: Thank you for your suggestion. All statistical analyses were performed using GraphPad Prism version 5.03 software. Cell counts, the expression of co-inhibitory molecules and cytokine/chemokines levels were compared by unpaired t test. The percent of T, B cell, CD4+ and CD8+ T cells, apoptosis and the mRNA expression of T, B and NK cells were analyzed by two-way ANOVA with Tukey’s multiple comparison test. Survival rates in mice were compared by the log-rank test. The related description has been added in section 2.9 and the figure captions of the revised version of the manuscript.

Figure captions for 1, 2, 3, 5, 6, 7, and 8: more specific methods of statistics should be mentioned.

Response: The more specific methods of statistic has been added in the caption 1, 2, 3, 5, 6, 7, 8, and 9 of the revised version of the manuscript.

Figure 2D: Statistical analysis is mandatory.

Response: Thank you for your suggestion. Statistical analysis of Figure 2D has been added in the revised version of the manuscript.

In figure 6, the authors showed IL-6 level was increased after 48 hours post-infection. In addition, in figure 2, decreased lymphocytes number were observed after 48 hours post-infection. If IL-10 is involved in the decreased lymphocytes number, the time gap among them can be observed.

Response: We are apologized that we had mistakenly not marked the significant analysis of the IL-6 in Figure 6. Although the level of IL-6 was increased after 48 hours post-infection, but the difference was not statistically significant between uninfected control and infected group. Therefore, it was speculated that IL-6 may be not involved in lymphopenia. In addition, the infection of FMDV induced the quick disease progression in mice and resulted in rapid death of all animals within 72 hpi. It was difficult to observe the time gap among the increase of IL-10 and the occurrence of lymphopenia when the level of IL-10 was detected every 12 hours. It was noted that the levels of IL-10 were coincided with the onset of lymphopenia, which supported that IL-10 was involved in lymphopenia.

Figure 7C: Why anti-IL-10R antibody treatment did not lower the level of apoptosis of B cells.

Response: In this study, knocking out IL-10 or blocking IL-10/IL-10R signaling in vivo were able to inhibit apoptosis of T cells, but not B cells. To best of our knowledge, IL-10 as an autocrine growth factor in malignant B-1 cells lies in its ability to inhibit apoptosis induction of B cells (Gary-Gouy et al., 2002; Yen Chong et al., 2001), which supported that anti-IL-10R antibody treatment may not lower the level of apoptosis of B cells. In contrast, IL-10 produced induced T cell apoptosis by extrinsic and intrinsic apoptosis pathways via a MAPK-dependent mechanism (Brosseau et al., 2018; Yang et al., 2015). Thus, knocking out IL-10 or blocking IL-10/IL-10R signaling in vivo was able to inhibit apoptosis of T cells, but not B cells. The related description has been added in the lines 763-766 of discussion section of the revised manuscript.

Figures 7 and 8: The statistics between mock and IL-10R groups are missing.

Response: Thank you for your suggestion. The statistics between mock and IL-10R/IL-10-/- groups have been added in figures 7 and 8 of the revised version of the manuscript.

Do you have any idea how the FMD virus induces IL-10 production?

Response: During FMDV infection, the elevated IL-10 level in serum has been widely reported in the serum of FMDV infected cattle and pigs (Díaz-San Segundo et al., 2009; Sei et al., 2016; Zhang et al., 2015). IL-10 is known to be produced by a variety of innate and adaptive immune cells, including macrophages, dendritic cells (DCs), natural killer (NK) cells, CD4, CD8, γδ T cells, and B cells (Rojas et al., 2017). To best of our knowledge, IL-10 have been reported to be produced by dendritic cells (DCs) and monocytes following FMDV infection (Sei et al., 2016; Díaz-San Segundo et al., 2009). However, it is still unknown that the mechanism of IL-10 production in DCs and monocytes during FMDV infections. We believed that it is interesting to further investigate the cell source of IL-10 during FMDV infection and how FMDV interacts with these cells to induces IL-10 production.

Minor comments:

“+” should be superscript, e.g., CD4+. Please carefully go through the manuscript and correct all of them.

Response: Thank you for your suggestion. It has been revised throughout the revised version of the manuscript.

IFN, IL, MHC, EDTA, NK, hpi, TNF, TGF, PBS, and ELISA should be spelled out at their first appearance. Please carefully go through the manuscript and correct all of them.

Response: Thank you for your suggestion. These abbreviations have been spelled out at their first appearance in the revised manuscript, and the abbreviation word in this manuscript has been listed in lines 29-37 of the revised version of the manuscript.

Page 7, line 294: Please correct “live”to “liver”.

Response: It has been revised in the revised version of the manuscript.

Round 2

Reviewer 1 Report

I commend the authors for taking the criticisms on the manuscript positively and making all efforts to incorporate the suggested changes.  The manuscript now reads well and is an important contribution  towards understanding the disease.  I would request the authors to make these three additional changes resulted due to the review 1.

Line 39: To start the manuscript please refer it to Foot-and-Mouth Disease (FMD).  I have debated with some of my learned colleagues in this field and the consensus was to use the term contagious rather than infections.  My apologies, please change it back to ‘contagious'.

Line 235-236: Change to ‘sera’ or ‘serum samples’

Line 142-143. We cannot characterize the disease in mice as they are not the target species for FMD.  We can only differentiate certain clinical signs post infection that are generally not expected in a normal mouse.  They can exhibit certain clinical signs due to infection but not specific to FMD.  Therefore, please change to ‘C57BL/6 mice infected with FMDV are expected to develop clear clinical signs of infection, including respiratory distress, neurological signs and wasting [27].’

Thank you and best wishes.

Reviewer 2 Report

I have gone through the revised manuscript, and I think all my comments have been adequately addressed.